# SUMO Interacting Motifs: Structure and Function

**DOI:** 10.3390/cells10112825

**Published:** 2021-10-21

**Authors:** Tak-Yu Yau, William Sander, Christian Eidson, Albert J. Courey

**Affiliations:** Department of Chemistry & Biochemistry, University of California, Los Angeles, CA 90095, USA; tomtakyu@chem.ucla.edu (T.-Y.Y.); wsander13@g.ucla.edu (W.S.); ceidson@g.ucla.edu (C.E.)

**Keywords:** SUMO, SUMO interacting motif, phase separation, DNA repair, host–pathogen interactions, histones, post-translational protein modification

## Abstract

Small ubiquitin-related modifier (SUMO) is a member of the ubiquitin-related protein family. SUMO modulates protein function through covalent conjugation to lysine residues in a large number of proteins. Once covalently conjugated to a protein, SUMO often regulates that protein’s function by recruiting other cellular proteins. Recruitment frequently involves a non-covalent interaction between SUMO and a SUMO-interacting motif (SIM) in the interacting protein. SIMs generally consist of a four-residue-long hydrophobic stretch of amino acids with aliphatic non-polar side chains flanked on one side by negatively charged amino acid residues. The SIM assumes an extended β-strand-like conformation and binds to a conserved hydrophobic groove in SUMO. In addition to hydrophobic interactions between the SIM non-polar core and hydrophobic residues in the groove, the negatively charged residues in the SIM make favorable electrostatic contacts with positively charged residues in and around the groove. The SIM/SUMO interaction can be regulated by the phosphorylation of residues adjacent to the SIM hydrophobic core, which provide additional negative charges for favorable electrostatic interaction with SUMO. The SUMO interactome consists of hundreds or perhaps thousands of SIM-containing proteins, but we do not fully understand how each SUMOylated protein selects the set of SIM-containing proteins appropriate to its function. SIM/SUMO interactions have critical functions in a large number of essential cellular processes including the formation of membraneless organelles by liquid–liquid phase separation, epigenetic regulation of transcription through histone modification, DNA repair, and a variety of host–pathogen interactions.

## 1. Introduction

Small ubiquitin-related modifier (SUMO) is a member of the ubiquitin-like protein family and, like ubiquitin, SUMO can be covalently attached to lysine side chains in a variety of target proteins. Unlike ubiquitylation, however, SUMOylation does not have a direct role in targeting proteins for degradation by the proteasome, and much research has been directed toward elucidating the consequences of this post-translational modification [1,2]. The SUMOylation pathway is similar to the ubiquitylation pathway and employs homologous enzymes. SUMO contains an extended C-terminal region which is cleaved by a member of the ubiquitin-related protease family, exposing a C-terminal di-glycine motif that is critical for covalent conjugation [3]. This mature form of SUMO is then attached to a SUMO-activating enzyme (a heterodimer composed of SAE1 and SAE2) via a thioester bond between the C-terminal carboxyl group on SUMO and a cysteine residue in the active site of SAE2. SUMO is subsequently transferred to an active site cysteine in the SUMO-conjugating enzyme (Ubc9), and finally, to a lysine residue in a target protein. Target protein selection often involves the action of SUMO ligases. SUMOylation is a dynamic process and can be readily reversed by SUMO proteases, which also function as SUMO deconjugating enzymes [4]. The SUMO acceptor lysine is often, though not always, embedded in a ѰKXE consensus motif, with Ѱ denoting a hydrophobic amino acid residue and X denoting any amino acid residue [5].

SUMO modification regulates a variety of biological processes [1,2,6,7]. Many of these functions rely on the ability of SUMO to promote non-covalent protein–protein interactions. This, in turn, depends on the non-covalent binding of covalently conjugated SUMO to SUMO interacting motifs (SIMs) in other proteins. This review will focus on the structural aspects of the SUMO/SIM interaction, the regulation of these interactions, and the roles of SIMs in a few biological processes.

It is likely that a large fraction, perhaps the majority, of SUMO functions require SIMs. Although this review will focus on processes in which the SIMs have been identified, it seems likely that many additional SUMO-mediated processes for which SIMs are a key player have yet to be identified. However, there may also be SIM-independent SUMO functions. For example, it is possible that, in some cases, covalent conjugation of SUMO to a protein alters its function by disrupting a protein-protein interaction or by directing changes in protein conformation (for examples, see [8,9]). 

This review is not meant to be exhaustive. There are many more examples of characterized SIMs than will be mentioned here. After reviewing what we know about SIM structure and function, we will describe a few of the many known processes in which SIMs play a critical role, including the formation of phase-separated membraneless organelles, histone function, DNA repair, and host–pathogen interactions. 

## 2. SIM Structure and Regulation of SIM/SUMO Complex Formation

### 2.1. The SIM Consensus

The initial discovery of SIMs resulted from a yeast two-hybrid screen for human proteins that interact with p73, which is a relative of the p53 tumor suppressor [10]. This screen isolated human SUMO-1 (one of five SUMO isoforms encoded in the human genome), the SUMO conjugating enzyme (Ubc9), and several SUMO-interacting proteins. Many of these SUMO interactors were found to contain a conserved motif containing a Ser-X-Ser sequence. However, an NMR chemical shift perturbation study [11] showed that the Ser-X-Ser motif was not required for binding to SUMO [although, as discussed below, it does play a role in the regulation of the interaction). Rather, binding requires a hydrophobic sequence adjacent to the Ser-X-Ser motif. This hydrophobic core is typically four amino acid residues in length and often matches either an hXhh consensus or an hhXh consensus, in which h is an amino acid with a large non-polar aliphatic side chain (I, V, or L) and X is any amino acid [12,13,14] (Figure 1A). This hydrophobic core is often flanked on one side or the other by negatively charged residues, i.e., aspartates, glutamates, and/or phosphoserines. Useful and empirically validated SIM prediction algorithms are based on this consensus [12,14]. Dissociation constants for SUMO/SIM complexes are generally in the micromolar or sub-micromolar range, and mutations that alter just one of the hydrophobic core amino acids can greatly destabilize binding [13,15,16].

It is worth nothing, however, that not all SIMs match the consensus described above. For example, Ssa1, the yeast orthologue of Hsp70, contains a SIM with the sequence TVEEVD, thus lacking a hydrophobic core with at least three non-polar aliphatic amino acids. However, this may not represent a truly distinct type of SIM, since it appears to bind to the same SUMO surface (to be discussed below) in much of the same manner as SIMs that follow the motif’s consensus more closely [18]. 

### 2.2. Structure of the SUMO/SIM Complex

SUMO and other ubiquitin-family proteins assume what has been termed a “β-grasp” fold, in which a four or five-stranded β sheet wraps around (grasps) an α helix [19]. The solution structure of a complex between the SIM in PIASX and human SUMO-1, as determined by NMR spectroscopy [13], demonstrates that the SIM binds in a hydrophobic groove between the second β strand and the α helix of SUMO (Figure 1B). The SIM assumes an extended conformation and the first several residues have dihedral angles consistent with the structure of a β strand, suggesting that the SIM and the second β strand of SUMO may interact through backbone hydrogen bonding interactions characteristic of β sheets [13]. However, hydrogen–deuterium exchange experiments do not show the slow exchange of the SIM backbone hydrogen atoms often found in β sheets, indicating that the backbone interactions are weak. Rather, the NMR data show that the SIM/SUMO complex is stabilized by hydrophobic side chain interactions between the residues in the SIM and those in the SUMO hydrophobic groove. In addition, the SUMO hydrophobic groove contains positively charged amino acid side chains, which permit favorable electrostatic interactions with negatively charged residues that often flank the hydrophobic core of a SIM; these electrostatic interactions increase the stability of the complex.

The solution structure of the complex between SUMO and the PIASX SIM confirms many of the features of the 3 Å-resolution X-ray crystal structure of the complex between SUMOylated RanGAP and the SIM in RanBP2, another SUMO ligase [20]. Both the PIASX and RanBP2 SIMs contain a core hydrophobic motif in an extended (β strand-like) conformation that makes non-polar contacts with the hydrophobic groove in SUMO. Surprisingly, however, the SIMs from these two proteins bind SUMO in opposite orientations—the PIASX SIM is parallel to the second SUMO β strand, while the RanBP2 SIM is antiparallel to the second SUMO β strand [13,20] (Figure 1B). 

It is possible that differences in the hydrophobic core are sufficient to determine whether binding occurs in the antiparallel or parallel orientation. For example, the PIASX SIM is of the type *hhXh*, whereas the RanBP2 SIM is of the type *hhhh*. It has been suggested [13] that the parallel binding by the PIASX SIM is preferred because the parallel orientation optimizes non-polar contacts between the *hhxh* SIM core and SUMO. This conclusion is challenged, however, by a molecular dynamics simulation of binding between the SIM in PIAS1 (a close relative of PIASX) and SUMO-3 [21]. This simulation suggests that the core binds with similar affinity to the SUMO hydrophobic groove in either the parallel or the antiparallel orientation. However, in the antiparallel complex, the residues flanking the SIM core are largely unstructured, while in the parallel complex, the flanking residues are constrained in favorable interactions with residues in SUMO. This implies that both binding affinity and binding orientation are strongly influenced by interactions involving residues outside the SIM hydrophobic core.

Other studies highlight the importance of the distribution of the flanking charged residues in determining the binding orientation. SIMs that bind to SUMO in the parallel orientation frequently contain a C-terminal negatively charged patch, while SIMs that bind in the antiparallel orientation frequently contain an N-terminal negatively charged patch [13,22,23]. In each case, the preferred orientation generally places the negatively charged patch in the SIM over a conserved constellation of positively charged residues in SUMO (Figure 1A,C).

The physiological significance of the SIM binding groove in SUMO is demonstrated by a study looking at the possible role of SUMO in the inhibition of transcriptional activation [24]. Many transcription factors are targets for covalent conjugation to SUMO. It is thought that this conjugation could, in many cases, lead to the recruitment of inhibitory co-regulators. Extensive mutagenesis of surface residues in both human SUMO-1 and human SUMO-2 revealed a set of mutations that prevented transcriptional inhibition by SUMO. These mutations map to the SIM binding pocket, including portions of the second β sheet and α helix of SUMO. 

The functional importance of the SIM binding interface in SUMO is also demonstrated by a high-throughput mutagenic screen for mutations in the gene encoding the sole SUMO family protein in yeast [25]. This study identified a number of lethal and conditional mutations in and around the SIM binding groove. For example, mutagenesis of two lysine residues in yeast SUMO thought to make electrostatic contact with negatively charged residues in SIMs was found to result in sensitivity to various forms of stress, including temperature stress and genotoxic stress

In conclusion, it is clear that SIMs typically contain an essential hydrophobic core of four amino acids that binds to a conserved hydrophobic pocket in SUMO and that a negatively charged stretch of amino acids often flanks this hydrophobic core. While we do not yet understand all the factors that influence binding orientation, it seems clear that electrostatic interactions involving the negatively charged residues play important roles in determining both affinity and orientation of the SUMO/SIM complex. 

### 2.3. Regulation of SUMO/SIM Complex Formation

SUMO/SIM complex formation is frequently regulated by post-translational modification. For example, phosphorylation of serine residues adjacent to the SIM hydrophobic core has been shown to stimulate binding to SUMO by introducing negative charges into the SIM that engage in favorable electrostatic interactions with positive charges in SUMO. The role of phosphorylation is particularly well-characterized in the case of the human Daxx protein [16]. Daxx is a transcriptional co-regulator with pro-apoptotic functions that interacts with the SUMO-conjugated form of the promyelocytic leukemia (PML) protein. Daxx and PML accumulate in the so-called PML nuclear bodies (PML-NBs), which are punctate phase separated bodies found in the nuclei of many cells (see the section below on role of SIMs in formation of phase separated organelles). The interaction of Daxx with SUMOylated proteins requires a SIM in Daxx with the sequence IIVLSDSD. Thus, its core hydrophobic sequence (IIVL) is immediately followed by a sequence of the type SXS, which, as mentioned above, was noticed as a characteristic of many of the earliest discovered SIMs. 

Both serine residues in the Daxx SIM are targets for phosphorylation by casein kinase II, and this phosphorylation appears to contribute to the pro-apoptotic function of Daxx. As shown by isothermal titration calorimetry, phosphorylation of one or both serine residues increases the affinity of the Daxx for SUMO-1 by up to 30-fold [16]. Determination of the structure of the complex by NMR suggests that this stabilization is due, in part, to an electrostatic interaction between the negatively charged phosphate groups and a particular lysine residue (K39) near the C-terminal end of the second SUMO β-strand. The structure of the complex between SUMO-1 and the Daxx SIM as determined by X-ray crystallography also supports the importance of these electrostatic interactions (Figure 1C). This structure shows that a number of positively charged residues in SUMO-1 (K37, K39, H43, and K46) provide a favorable electrostatic environment for the negative charges introduced as a result of phosphorylation of the Daxx SIM [17]. Phosphorylation of Daxx only stimulates binding to SUMO-1 and not binding to SUMO-2 or SUMO-3 [16]. It is possible that this difference can be accounted for, at least in part, by the presence of His43 in SUMO-1, a residue that is not conserved in SUMO-2 or SUMO-3. The crystal structure of the SUMO-1/Daxx SIM complex shows a tight interaction between the second phosphoserine in Daxx and this histidine side chain (Figure 1C).

Post-translational modification of SUMO may also function to modulate the affinity of the SIM/SUMO complex. Several of the lysine residues in the SIM binding groove of SUMO are targets for acetylation, which could modulate SIM affinity. This is demonstrated by studies of PIAS family SUMO ligases [26]. PIAS1, PIAS2, and PIAS3 each contain two SIMs; both SIMs contain one or more negatively charged residues (aspartates or phosphoserines) C-terminal to the hydrophobic core. Structural studies demonstrate electrostatic contacts between these negatively charged residues and lysines in the SUMO groove. Furthermore, acetylation mimetic substitutions (lysine to glutamine) in several of the lysine residues significantly reduce the stability of this complex. 

The consequences of lysine acetylation and how lysine acetylation interacts with serine phosphorylation have been further illuminated through studies of the effect of SUMO acetylation upon the binding of the phosphorylated forms of the SIMs in Daxx and PML [27]. Binding to both phosphoSIMs is greatly attenuated by acetylation of either K39 or K46 in SUMO-1. In contrast, acetylation of K37 interferes with SUMO binding to Daxx, but not with the binding to PML. X-ray crystallographic analysis shows that, in both cases, acetylation of SUMO K37 results in changes in the SUMO/SIM interface. In the complex between SUMO and the Daxx phosphoSIM, these changes lead to a decrease in favorable electrostatic contacts, thus reducing complex stability. However, in the case of the complex involving the PML phosphoSIM, compensatory changes in electrostatic contacts may cancel one another out. This demonstrates that the plasticity of the SUMO/SIM interface may lead to SIM-specific effects of SUMO acetylation on SIM binding.

## 3. Specificity and the SUMO Interactome

Cells contain a wealth of proteins that interact non-covalently with SUMO. A systematic effort to identify the non-covalent SUMO interactome has been carried out. This study involved the use of SUMO-1, SUMO-2, and a SUMO-2 trimer as affinity reagents to purify interacting proteins from HeLa cell lysates [28]. In this way, 379 interacting proteins were identified. Many of the identified proteins bind only to SUMO-2 or the SUMO-2 trimer, whereas a much smaller number bind only to SUMO-1. Other interactors bind with similar affinity to SUMO-1 and SUMO-2. A search of these non-covalent SUMO-interacting proteins for sequences with similarity to the SIM consensus motif turned up over 800 putative SIMs. Given the degenerate nature of the SIM consensus, it is likely that many of these SIMs do not represent true biologically relevant SUMO interaction motifs. However, eight of the putative SIMs from proteins that selectively bind SUMO-2 were validated as true SUMO-2 binding SIMs in an ELISA assay. In each case, binding was substantially diminished by mutagenesis of core hydrophobic amino acids. 

Given the large number of SUMO-interacting proteins, it is essential to determine the basis for specificity in SIM/SUMO interactions, an issue that remains largely unaddressed. SUMOylation has been implicated in a broad range of biological activities including pre-mRNA splicing, transcription, mitosis, protein stabilization, and DNA repair [1,2]. Logically, each SUMOylated protein must interact with particular SIM-containing targets appropriate to its function. Although we do not fully understand the basis for this selectivity, a few possibilities suggested by clues in the scientific literature are discussed below.

One possible explanation for selectivity in SUMO/SIM interactions is that SUMOylated proteins do not solely contact the SIM in the target protein, but also interact with other surfaces in the target protein. These additional binding surfaces could function as determinants of specificity by mediating the formation of a cooperative complex that requires the SUMO/SIM interaction and one or more additional protein-protein interactions. An example of such a complex is provided by the ternary complex of SUMO-conjugated RanGAP1 with Ubc9 (the SUMO-conjugating enzyme) and RanBP2 (a SIM-containing SUMO ligase). In this complex, RanBP2 contacts SUMO via its SIM and simultaneously makes multiple stabilizing contacts with Ubc9 [20]. A second example of a SUMO/SIM-containing complex that is stabilized by additional interactions outside SUMO and a SIM is provided by the complex that includes SUMOylated PCNA and Elg1, which is discussed below in the section on SIMs in DNA repair.

In other cases, a degree of specificity may be achieved through poly-SUMOylation, in which a target lysine is attached to a chain of SUMO proteins. Poly-SUMOylation can occur because some SUMO isoforms contain internal SUMO acceptor lysine residues that allow the attachment of one SUMO to another, which makes the formation of a poly-SUMO chain possible [29]. Some SIM-containing proteins specifically recognize poly-SUMOylated proteins. This includes a family of RING-domain containing ubiquitin ligases termed SUMO-targeted ubiquitin ligases (STUbLs). These proteins, which include human RNF4 and RNF111, as well as Drosophila Degringolade, contain multiple SIMs [30,31,32,33,34]. This allows them to bind poly-SUMOylated proteins and then use their RING domains to catalyze ubiquitylation of the bound protein, altering protein function in a variety of ways. Although not proven, it seems likely that the multiple SIMs in STUbLs simultaneously engage the multiple SUMO moieties in a poly-SUMO chain, thus explaining why they are specific for poly-SUMOylated targets [33,35]. 

Yet, another mechanism for achieving specificity in SUMO/SIM interactions could be SUMO isoform-specific SIMs. As mentioned above, a proteomic study has revealed 379 SUMO interacting proteins, many of which have differential affinity for SUMO-1 and SUMO-2 [28]. Gene ontology analysis of these proteins shows that, while regulators of protein SUMOylation and RNA processing bind similarly to SUMO-1 and SUMO-2, DNA damage response proteins are significantly enriched for SUMO-2-interacting proteins relative to SUMO1-interacting proteins. Thus, different SUMO isoforms could direct different biological processes. 

The finding that SIMs can be SUMO isoform-specific is not surprising given that human SUMO-1 and human SUMO-2 share only 44% sequence identity. The possibility of SUMO isoform-specific SIMs has also been confirmed by a study in which a phage display screen was used to identify sequences with differential affinity for SUMO-1 and SUMO-2 [36]. Structural analysis and molecular dynamics suggest that preferential binding of a SIM to one SUMO isoform or another is due in part to hydrogen bonding interactions. However, it has not been possible to formulate simple rules that would allow one to distinguish SUMO-1 and SUMO-2-binding SIMs from one another by inspection of their sequences. Given the small number of isoforms, isoform specificity is not likely to account for most of the diversity in SIM function and would not explain how organisms such as budding yeast or fruit flies, with only a single SUMO isoform each, are able to achieve SUMO/SIM functional diversity.

## 4. Role for SUMO/SIM Interactions in Structure and Function of PML Bodies and Other Phase Separated Organelles

SUMO/SIM interactions may play a central role in the formation of phase-separated membraneless organelles. These organelles are protein aggregates that form spherical droplets within the cytoplasm or nucleoplasm in a phenomenon known as liquid–liquid phase separation (LLPS) [37]. It is possible to create phase-separated droplets in vivo by mixing a peptide containing multiple copies of the PIASX SIM (poly-SIM) and a peptide containing multiple repeats of human SUMO-3 (poly-SUMO) [37]. This leads to the formation of poly-SIM/poly-SUMO-containing droplets visible under a light microscope.

One of the oldest discoveries pertaining to SUMO is the observation that it is enriched in a membraneless nuclear organelle known as the PML nuclear body (PML-NB), which is found in many types of mammalian cells [6,38,39]. These organelles have been shown to have antiviral capabilities, among other functions.

PML-NBs contain the promyelocytic leukemia (PML) protein, which is highly SUMOylated and contains a SIM [6]. Interactions between the PML SIM and SUMOylated proteins including PML and DAXX mediate the formation of the PML-NBs. The SIM in PML consists of a four-amino acid hydrophobic core (VVVI), which is followed by a stretch of amino acids rich in serines and acidic residues. The serine residues can be phosphorylated, and this phosphorylation increases the affinity of PML for SUMO (especially SUMO-1) by strengthening the electrostatic interactions with positively charged residues in SUMO [17].

One role of SUMO/SIM interactions in the function of PML nuclear bodies has been demonstrated in studies of cancer cell proliferation. The unchecked growth of cancer cells requires a mechanism to avoid the excessive shortening of telomeres that occurs in normal cells at every mitotic cycle, which would normally limit the number of cell cycles before permanent arrest of the cell cycle (replicative senescence) or apoptosis [40]. Telomere maintenance usually involves the action of a reverse transcriptase called telomerase, but some cancers employ a telomerase-independent mechanism for maintaining telomere length called “alternative lengthening of telomeres” (ALT) [41]. ALT has been shown to work through PML-NBs. In cells undergoing ALT, telomeres cluster in PML-NBs to form ALT-associated PML-NBs (APBs). Although PML-NBs generally dissolve during mitosis, APBs persist, possibly as a consequence of their hyper-SUMOylated state.

In an effort to determine if APBs are functionally required for ALT, artificial Poly-SIM/Poly-SUMO aggregates were targeted to the nuclei of human embryonic kidney 293 cells by the addition of a nuclear localization signal to Poly-SIM [42]. Poly-SIM was also engineered to contain a protein domain from the transcription factor RAP1 that is known to bind telomeres with high affinity. This led to the formation of phase-separated aggregates in the nucleus. Telomeres clustered to these aggregates, mimicking the ALT associated PML-NBs found in some cancers. This by itself was not sufficient to induce telomere lengthening, which additionally required the overexpression of BLM, a helicase with roles in many processes including homologous recombination, DNA double stranded break repair, and the ALT process described here [42]. BLM contains two SIMs and mutagenesis of these SIMs resulted in the failure of BLM to be recruited to the poly-SUMO/poly-SIM scaffold and, therefore, to the loss of telomere lengthening.

A second, and perhaps even more remarkable example of how SIM/SUMO interactions in PML-NBs bring about a complex biological response, involves ICP0, a STUbL encoded in the Herpes Simplex Virus 1 (HSV-1) genome. This protein has a central role in the interplay between the host cell’s antiviral response and the attempt of the virus to circumvent such a response to further its own proliferation [43]. It will be discussed in detail in the section below on the role of SIMs in host–pathogen interactions. 

The role of SUMO/SIM interactions in PML-NB formation may be but one example of a broader role of these interactions in the formation of phase-separated membraneless organelles. For example, these interactions could drive or regulate the formation and/or dynamics of nuclear speckles, which are phase-separated ribonucleoprotein aggregates with essential roles in RNA processing [44]. In addition, the Polycomb bodies found in many metazoan cell types are enriched for SUMO [45]. 

More generally, it appears that SUMO modification of multiple members of large protein groups, such as those involved in the formation of phase-separated bodies and inDNA repair, may serve as a strategy for the synergistic assembly of protein complexes [46,47]. This is achieved through the presence of SIMs in these SUMO acceptor proteins that mediate the formation of multiple additive or greater than additive protein-protein interactions, resulting in the formation of large stable complexes. 

## 5. Role of SUMO/SIM Interactions in Histone Function

Most core histones, linker histones, and histone variants are targets of SUMOylation [48]. SUMOylation of histones often leads to transcriptional repression either by antagonizing activating histone marks or by directly recruiting regulators with repressive activity such as histone deacetylases. Histone modification can promote the recruitment of negative regulators and this process sometimes requires SUMO/SIM interactions (reviewed in [49]). In this section, we will describe two examples of this phenomenon: one involving the SUMOylation of histone H2B [50], and another involving the SUMOylation of histone H4 [51].

### 5.1. Recruitment of a SIM Containing Histone Deacetylase Complex

SUMOylation of histone H2B in yeast is stimulated by the dimethylation of histone H3 lysine 4, a histone mark associated with the 5′ ends of actively transcribed genes. SUMOylated histone H2B, in turn, recruits the Set3 histone deacetylase complex (HDAC) leading to deacetylation of histone H3 [50] (Figure 2A). In support of this sequence of events, a mutant form of histone H2B lacking four SUMO-acceptor lysines resulted in decreased recruitment of the Set3 HDAC and, as a consequence, histone H3 deacetylation did not occur. 

Recruitment of the Set3 complex by SUMOylated histone H2B is mediated by a SIM in Cpr1 (core sequence IVVA), which is a subunit of the Set3 complex [50]. Mutation of this SIM reduced recruitment of the Set3 complex to chromatin, an effect that was exacerbated further if the SIM mutation was combined with mutations in the histone H2B SUMO acceptor lysines.

Since both SUMO and histone deacetylation are generally associated with transcriptional repression, it seems paradoxical that histone H2B SUMOylation is enriched at transcriptionally active genes [50]. The resolution of this conflict may lie in the discovery that H2B SUMOylation and consequent histone H3 deacetylation could serve to inhibit the spurious transcription of non-coding RNAs that initiate within mRNA transcription units and interfere with productive elongation through these transcription units.

### 5.2. SIM-Mediated Crosstalk between Histone H3 and Histone H4

Further evidence for SIM-mediated crosstalk between SUMOylated histones and components of the histone modification machinery is provided by a biochemical study examining nucleosomes in a cell-free system [51]. Lysine specific demethylase 1 [LSD1) is a histone demethylase that forms a complex with co-repressor for element 1 silencing transcription factor (CoREST). LSD1 functions to silence transcription by demethylating mono- and dimethylated lysine 4 of histone H3 [52].

The discovery of a SIM in CoREST prompted an investigation into the possibility that histone demethylation might be regulated by histone SUMOylation [51]. This study utilized a strategy employing protein ligation involving chemically modified synthetic peptides to synthesize milligram amounts of histone H4 that is quantitatively SUMOylated on lysine 12 and histone H3 that is quantitatively dimethylated on lysine 4. These modified histones (with unmodified histones serving as the controls) were then incorporated into nucleosomal arrays, which were subjected to possible demethylation by the LSD1–CoREST complex. SUMOylated histone H4 was found to stimulate demethylation of histone H3 in these assays (Figure 2B). This effect depended on the SIM in Co-REST as a triple mutation in the SIM abrogated the effect. This stimulation of demethylation was limited to histone H3 polypeptides that were part of the same nucleosome as the SUMOylated histone H4 and did not extend to adjacent unSUMOylated nucleosomes. That is, the stimulation of demethylation was strictly intranucleosomal, which suggests that this particular type of histone SUMOylation does not mediate the spread of an altered chromosomal state, as is thought to be required for some epigenetic phenomena.

It is worth noting that some of the effects of SUMO on chromatin structure and function may be independent of SIMs. For example, studies with semi-synthetic nucleosomal arrays suggest that SUMOylation of histone H4 on lysine 12 could alter chromatin compaction by disrupting long-range internucleosomal interactions [8].

## 6. Roles for SUMO/SIM Interactions in Double Stranded DNA Break Repair and Nucleotide Excision Repair

SUMO/SIM interactions play a prominent role in DNA repair, including the repair of double stranded DNA breaks as well as nucleotide excision repair. Proliferating cell nuclear antigen (PCNA) is a SUMOylated protein that is central to the repair of double stranded DNA breaks. This protein forms a trimeric “sliding clamp” around the DNA and associates with DNA polymerases to greatly increase their processivity. PCNA also serves as an interaction scaffold for DNA replication and repair factors [53]. In yeast, SUMOylation of PCNA on lysines 127 and 164 occurs primarily during S phase or in response to DNA damage [54].

### 6.1. SIM-Mediated Interactions between PCNA, Elg1, and Srs2 in the Salvage Repair Pathway

Loading and unloading of PCNA on DNA requires a protein complex known as Replications Factor C (RFC) [55]. Elg1 is a yeast protein with sequence similarity to the largest subunit of RFC. It associates with additional polypeptides to form an RFC-like complex that increases the stability of the yeast genome [56]. The absence of this complex leads to chromosomal breaks, fusions, and inversions. Elg1 binds to both SUMOylated and unSUMOylated PCNA, but SUMOylation enhances the interaction significantly. This enhanced interaction is due to the presence of three SIMs in the N-terminal region of Elg1 (Figure 3A). The three SIMs function in an additive manner to mediate binding to SUMO, thus directing association of the RFC-like complex with PCNA [57]. 

Interestingly, Elg1 also contains a PCNA interacting protein (PIP) motif that binds directly to PCNA [57]. This suggests that binding of Elg1 to SUMOylated PCNA could depend on synergistic binding to SUMO through the SIMs and to PCNA itself through the PIP motif. This kind of synergistic or cooperative binding could explain how SIM-containing proteins bind specifically to one SUMOylated protein and not others.

Deletion of Elg1′s SIM and PIP motif-containing region results in the accumulation of SUMOylated-PCNA on chromatin, which suggests that the Elg1-containing RFC-like complex normally functions to unload SUMOylated-PCNA from chromatin (Figure 3B). The failure to unload SUMOylated PCNA in the absence of functional Elg1 leads to genomic instability. This is thought to reflect the function of Srs2, another SIM-containing factor [58]. During normal DNA replication, Srs2 associates with SUMOylated PCNA and prevents excessive homologous recombination by displacing homologous recombination proteins such as Rad51 from the DNA (Figure 3C). In addition to blocking homologous recombination, Srs2 also blocks a DNA repair pathway termed “salvage recombination” in which the sister chromatid is used as a template to replicate past DNA lesions. Thus, accumulation of excess SUMOylated PCNA on the DNA leads to excessive accumulation of Srs2, which leads to the inhibition of salvage recombination and an increased sensitivity to DNA damage.

### 6.2. SIMs in Nucleotide Excision Repair

Nucleotide excision repair (NER) is a DNA repair pathway responsible for repairing DNA distorting lesions such as UV-induced pyrimidine dimers and other pyrimidine photo adducts [59]. Many of the factors mediating this pathway were discovered through genetic analysis of Xeroderma Pigmentosum (XP), a human condition associated with extreme sensitivity to UV radiation. One of these factors, XPC, binds DNA lesions and recruits additional repair factors.

Poly-SUMOylation of XPC increases upon exposure to UV radiation [60]. Poly-SUMOylated XPC is subsequently recognized by RNF111/Arkadia, a STUbL, which employs its three adjacent SIMs for recognition of poly-SUMO-2 chains. RNF111 then mediates the ubiquitylation of XPC, which is required for its recruitment to sites of UV-induced DNA damage [32] (Figure 3D). In accord with this model, mutagenesis of the SIMs in RNF111 interferes with NER by preventing RNF111 from binding and ubiquitylating XPC.

## 7. SUMO and SIM in Pathogen–Host Cell Interactions

SUMO/SIM interactions are a key feature of the interplay between many pathogens and their hosts. The pathogens that manipulate SUMO are diverse, and examples are found among fungi, bacteria, and viruses. On one hand, SUMO and SIM can be used by the host cell to enhance cellular defenses against infection. On the other hand, the pathogens can hijack the SUMO pathway to aid in their proliferation or to overcome host cell defenses. Therefore, SUMOylation and non-covalent SUMO interactions mediate the struggles between a diverse set of host cells and their pathogenic adversaries. This section presents a few of the better characterized examples in which the mechanisms are beginning to be illuminated.

### 7.1. A SIM/SUMO Interaction in the Response to H. pylori Infection

A relatively straightforward example of how a SUMO/SIM interaction aids in the fight against infection is provided by the response of gastric cells to infection by the bacterium *Helicobacter pylori* [61], which is a twisted torpedo-shaped Gram-negative bacteria propelled by flagella [62]. Although many infected individuals are asymptomatic, *H. pylori* can attack gastric epithelial cells, potentially leading to disease [63]. The *H. pylori* enzyme urease decreases the acidity of the stomach, leading to inflammation, gastritis, and even stomach ulcers in the host (reviewed in [64]).

In response to infection, gastric cells limit the spread of *H. pylori* by triggering apoptosis (programmed cell death) in a manner that requires the nuclear localization of the p38 MAP kinase [61]. This kinase lacks a canonical nuclear localization signal (NLS) and is localized to the cytoplasm prior to infection. Upon infection, the expression of SUMO is upregulated. SUMO-2 then binds to p38, and this binding leads to p38′s nuclear import. The nuclear import of p38 is strongly dependent upon a SIM near the C-terminus of p38 (core sequence: LVLD). Free SUMO is largely nuclear, and so it is possible that the SUMO-2-dependent nuclear import of p38 reflects non-covalent binding of p38 to free SUMO. Alternatively, p38 may bind, via its SIM, to a currently unidentified SUMO-conjugated protein containing a canonical NLS. The exact details of this novel nuclear import pathway remain to be determined.

### 7.2. SIM/SUMO Interactions in Jasmonic Acid Signaling in Plants

A second, more complex example of a SIM/SUMO interaction in a pathogen response is provided by the Jasmonic Acid (JA) pathway in plants. The JA pathway regulates defense against necrotrophic pathogens (pathogens that kill a host cell before consuming it) such as the fungus *B. cinerea* [65,66]. In the absence of a hormonal signal, JAZ transcriptional co-repressors such as JAZ6 block the expression of plant genes involved in the defense against these pathogens [66]. In response to infection, JA is conjugated to isoleucine and the resulting JA-Ile hormone binds and activates the JA-Ile receptor COL1. In addition to being the JA-Ile receptor, COL1 is the subunit of a ubiquitin ligase complex that, when activated by its ligand, targets this complex to JAZ family co-repressors. The resulting ubiquitylation of the co-repressors leads to their proteasome-dependent degradation, thereby allowing expression of the antipathogen genes. 

A hint that the SUMO pathway may regulate this response comes from the following observations [66]. First, mutagenesis of the *A. thaliana* genes encoding the SUMO deconjugating enzymes OTS1 and OTS2 results in high levels of SUMOylated JAZ6, which increases JAZ6 stability by preventing COL1-mediated degradation of JAZ6. This, in turn, interferes with the activation of antipathogen genes in response to the JA signal, rendering the host more susceptible to *B. cinerea*. Second, *B. cinerea* infection triggers the degradation of OTS1 and OTS2. Thus, it is possible that rapid degradation of SUMO deconjugating enzymes in response to infection allows for the rapid activation of the antipathogen response even before sufficient levels of the JA-Ile conjugate have built up to trigger this response.

The mechanism by which SUMO conjugation to JAZ6 interferes with COL1-mediated degradation is not clear, but it appears to require binding of the SUMO attached to JAZ6 to an evolutionarily conserved SIM in COL1 (core sequence: VPEV) [66]. This might be expected to trigger COL1-mediated degradation of JAZ6 by targeting the ubiquitin ligase to its substrate. Ironically, it does just the opposite, i.e., it prevents JAZ6 ubiquitylation and degradation. It is not clear how this happens, but it is possible that COL1 cannot engage the degron in JAZ6 while its SIM is bound to a SUMO moiety that is covalently conjugated to JAZ6. Note that, like STUbLs, the COL1 ubiquitin ligase contains a SIM that critically regulates its function. In this case, however, the SIM somehow interferes with ubiquitylation of a SUMOylated target, whereas the SIMs in STUbLs promote the ubiquitylation of SUMOylated targets.

### 7.3. Regulation of Viral Pathogenicity by SIMs

Viral pathogens can also employ SIMs as a way of promoting their own survival and proliferation inside the host. A good example of this concept is provided by Kaposi’s sarcoma-associated herpesvirus (KSHV), the genome of which encodes both a SUMO ligase and a STUbL. Like many SUMO ligases, the KSHV SUMO ligase, K-bZIP, contains a SIM (core sequence: VIDL) [15]. This SIM binds to SUMO-2 with a sub-micromolar dissociation constant, and about ten-fold more weakly to SUMO-1. K-bZIP mediates its own SUMOylation as well as that of a variety of host cell proteins. One such host protein is the histone demethylase JMJD2A [15]. SUMOylation of this enzyme increases its affinity for euchromatic domains in the viral genome where it reactivates the KSHV latent region to restart the viral lytic cycle [15,67]. In addition, the SIM in K-bZIP is required for the SUMOylation of tumor suppressors p53 and Rb. This inhibits apoptosis, thus promoting proliferation of the virus inside the host [15]. 

Both KSHV and herpes simplex virus-1 (HSV-1) encode STUbLs that allow these viruses to circumvent host cell defenses conferred by PML-NBs [43,68]. When HSV-1 enters the host nucleus, the viral genome can be bound and silenced by PML-NBs [43]. Robust viral proliferation and lytic cycle reactivation require the action of ICP0, a STUbL encoded in the viral genome. This protein contains a SIM that, in its unmodified form, only binds to SUMO with low affinity. However, the phosphorylation of two serine residues near the SIM by a host cell kinase increases the ability of the SIM to bind SUMOylated PML-NB proteins, including PML itself. This leads to ICP0-mediated polyubiquitylation of SUMOylated proteins in the PML-NBs, targeting them for degradation by the proteasome, which is required to promote viral proliferation and lytic reactivation within the host cell [43] (Figure 4C). The mechanism for HSV-1 activation depicted here is striking, as it involves the interplay between three post-translational modifications: SUMOylation, ubiquitylation, and phosphorylation.

## 8. Concluding Remarks

In some ways, SUMOylation is like any reversible post-translational protein modification, in which a covalent change to protein structure serves as an on–off switch that alters protein function. However, unlike modifications such as phosphorylation or acetylation, which involve the addition of a small chemical group to a protein, SUMOylation involves the attachment of one entire protein to another. This added protein can then serve as a platform for the recruitment of diverse SUMO-interacting proteins through SUMO/SIM interactions. SUMO family proteins have evolved a hydrophobic groove that is dedicated to these interactions and which presumably dictates SUMO isoform specificity. This SIM-interacting groove is no doubt part of what differentiates SUMO from other ubiquitin-family proteins, such as ubiquitin or Nedd8.

One of the biggest gaps in our understanding of SIM function is the mechanism of interaction specificity. What is it that determines which of the hundreds or thousands of SIM-containing proteins in a proteome will interact with any given SUMOylated protein? This must, at least in part, result from cooperative interactions involving interactions between regions of the SUMOylated protein and the SIM-containing protein outside SUMO and the SIM. Although we have presented a few illustrative examples of this concept, in most cases, the details remain to be determined. 

Another feature of SUMO that differentiates it from other post-translational protein modifications is the possibility of multivalent interactions that can occur when a protein such as PML is both SUMO-conjugated and able to interact non-covalently with SUMO through a SIM. This can lead to the formation of large protein aggregates resulting in the genesis of membraneless organelles such as PML-NBs via the phenomenon of liquid–liquid phase separation. A number of functions of SUMO (e.g., in ALT and in viral de-toxification) are already known to require these organelles and it seems likely that they will be found to have, as of yet, unappreciated roles in other SUMO functions.

## Figures and Tables

**Figure 1 cells-10-02825-f001:**
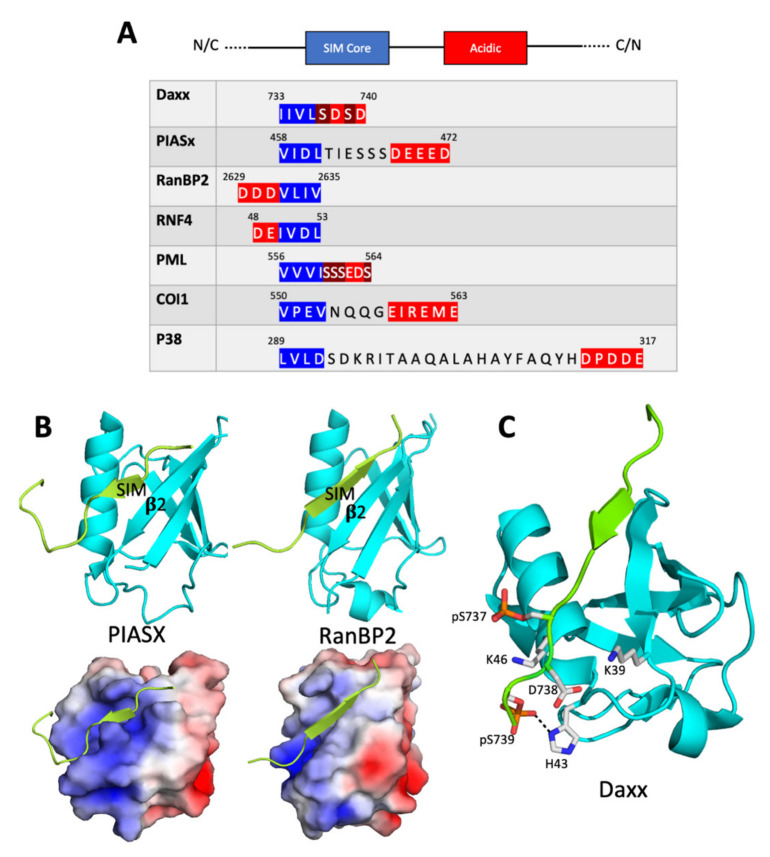
SIM consensus and the structure of the SUMO/SIM complex. (**A**) The consensus SIM structure which includes a SIM hydrophobic core flanked by negatively charged residues is shown at the top. The consensus can run in either direction (N-terminal to C-terminal or C-terminal to N-terminal). Below the consensus structure are portions of sequences from a few of the SIM-containing proteins discussed in the text. The SIM hydrophobic cores (blue) and negatively charged regions (red) are highlighted. Two serine residues in Daxx and four serine residues in PML that are phosphorylated to increase the affinity of the SIM for SUMO are highlighted in brown [17]. (**B**) The structures of the PIASX SIM bound to human SUMO-1 (left, PDB ID 2ASQ) and the RanBP2 SIM bound to human SUMO-2 (right, PDB ID 3UIN). In the cartoon representations at the top, the SIMs are green and the SUMOs are cyan. The second β strand in each SUMO is labeled. In the bottom images, the SUMOs are represented as electrostatic surfaces, with blue indicating positive charge and red indicating negative charge. In the PIASX complex, the SIM is parallel to the second β strand, while in the RanBP2 complex, the SIM is antiparallel to the second β strand. (**C**) The structure of the phosphorylated Daxx SIM bound to SUMO-1 (PDB ID 4WJP). In these cartoon representations, SUMO-1 is shown in cyan and the Daxx SIM is shown in green. Side chains of three negatively charged Daxx SIM residues (phospho-Ser737, Asp738, and phospho-Ser739) and three positively charged SUMO-1 residues (Lys39, His43, Lys 46) are shown. A dashed line indicates a close contact between phospho-Ser739 and His43. The electrostatic attractions between the positively charged SUMO residues and the negatively charged SIM residues serve to direct binding of the SIM in the parallel orientation.

**Figure 2 cells-10-02825-f002:**
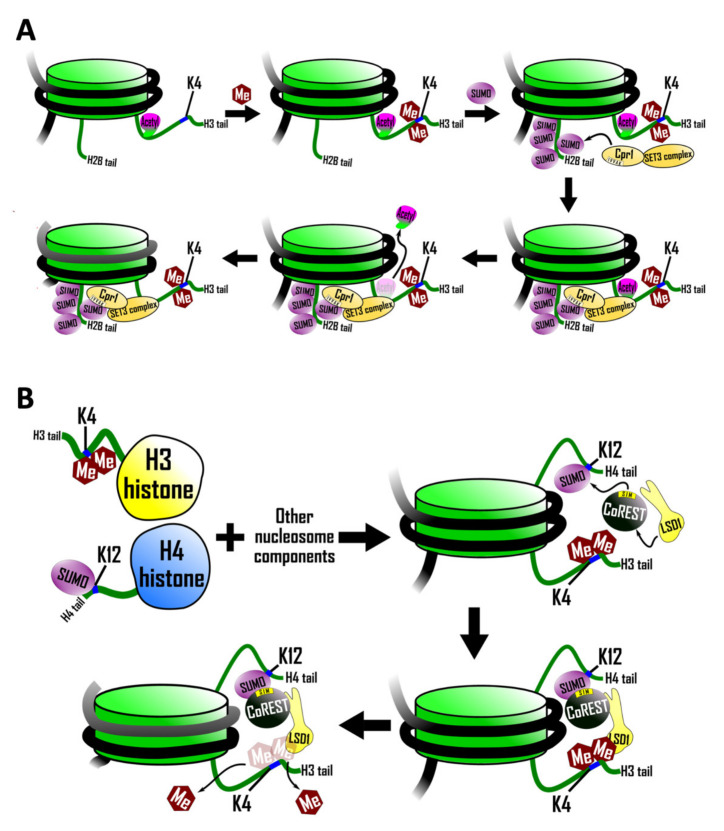
Regulation of histone modification by SUMO/SIM interactions. (**A**) Histone H3 is dimethylated on K4 (by Set1). Dimethlyated H3 histone then promotes the SUMOylation of histone H2B. Next, SUMOylated histone H2B recruits the Set3 histone deacetylase complex (HDAC) by interaction with a SIM in Cpr1, a subunit in the Set3 complex. Finally, Set3 removes acetyl groups from H3 to repress transcription. (**B**) K4-dimethylated histone H3 and K12-SUMOylated H4 are combined with DNA to form nucleosomes in vitro. Nucleosomes are then incubated with CoREST and LSD1. LSD1 and CoREST are only able to increase the rate of de-methylation of histone H3 K4 when the SIM on CoREST is intact.

**Figure 3 cells-10-02825-f003:**
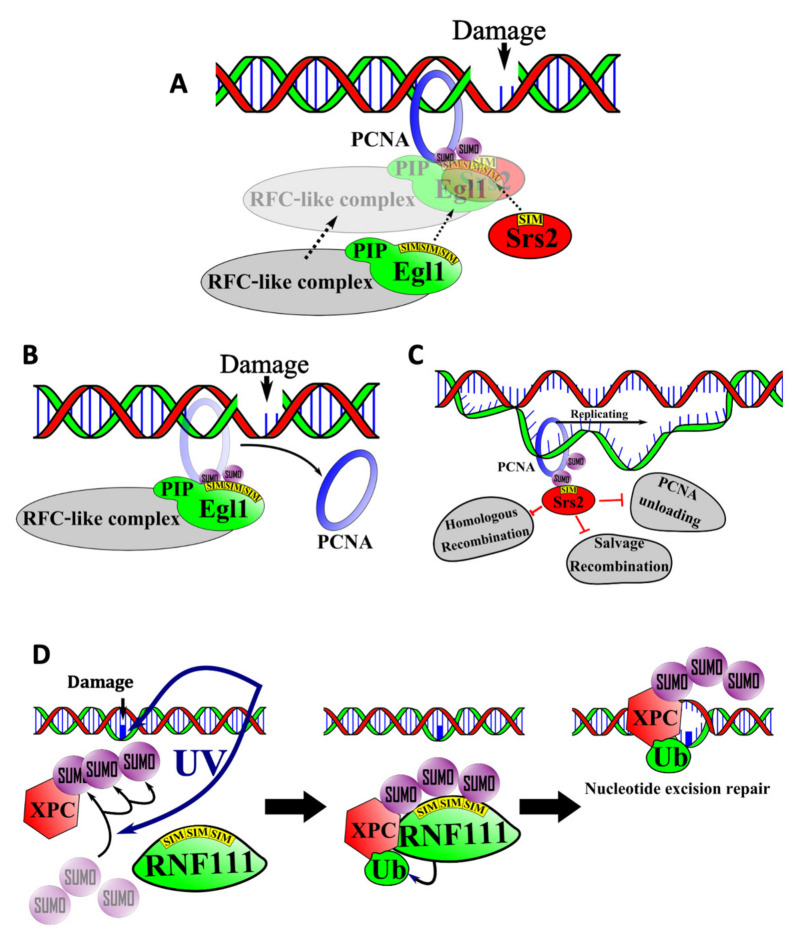
Roles for SUMO/SIM interactions in DNA repair. (**A**) The Elg1/RFC-like complex SIMs and the Srs2 SIM compete for binding to SUMOylated PCNA. (**B**) When the Elg1/RFC-like complex SIM interacts with SUMOylated PCNA, the PCNA is unloaded from the DNA, preventing accumulation of PCNA on the DNA and further DNA damage. (**C**) When the Srs2 SIM interacts with SUMOylated PCNA, the interaction inhibits the Elg1-mediated unloading of PCNA, homologous recombination, and the salvage recombination pathway. (**D**) The SIMs in RNF111, a SUMO-targeted ubiquitin ligase, interact with SUMOylated XPC to catalyze its nonproteolytic ubiquitylation, which activates the nucleotide excision repair pathway.

**Figure 4 cells-10-02825-f004:**
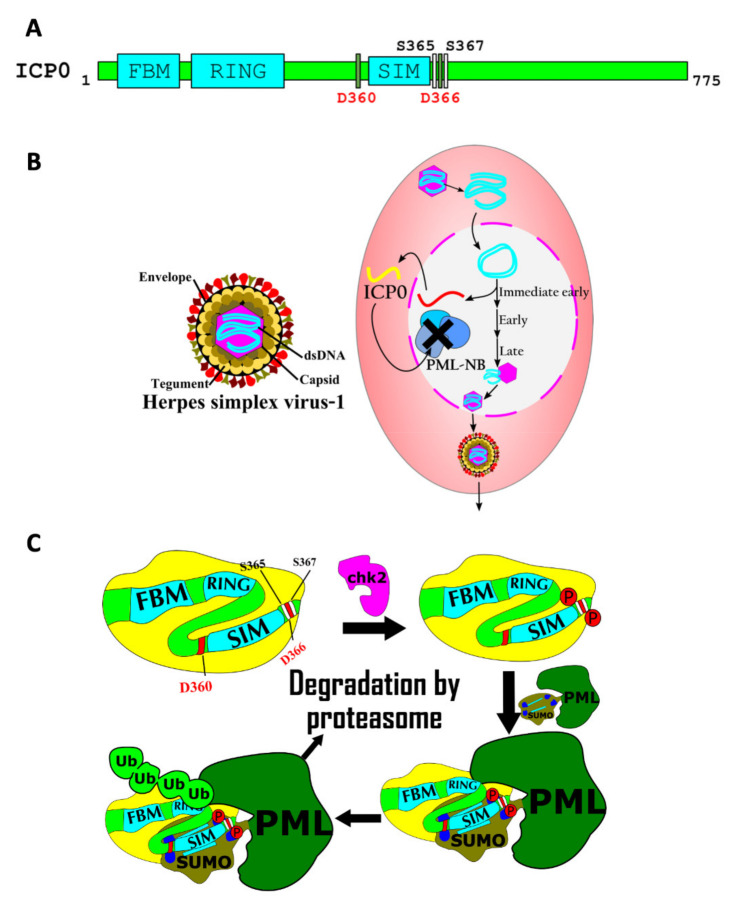
A role for a SIM in the viral ICP0 protein in HSV-1 host–pathogen interaction. (**A**) The primary structure of Herpes simplex virus 1 ICP0 protein, a SUMO-targeted ubiquitin ligase. The FHA-binding motif (FBM), the RING finger ubiquitin ligase domain (RING), and the SUMO interacting motif (SIM) are shown. Two aspartate residues and two serine residues that modulate the interaction between SUMO and the SIM are labeled. Phosphorylation of the serine residues greatly increases the affinity of the SUMO/SIM interaction. (**B**) The HSV-1 virion is depicted on the left. A simplified viral infectious cycle is displayed on the right. The DNA genome of HSV-1 is rapidly circularized in the nucleus of a human epithelial cell. ICP0 is an immediate-early (IE) gene that is expressed to direct the degradation of host cell PML protein within the host cell, which would otherwise inactivate the viral genome as a component of the PML nuclear bodies (PML-NB). (**C**) The ICP0 STUbL protein is phosphorylated by chk2 protein kinase. Phosphorylation of ICP0 increases the affinity of its SIM for SUMOylated PML proteins. Finally, the RING domain of ICP0 poly-ubiquitylates PML to target PML for degradation, thus promoting viral proliferation.

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
