# Peer review of "SUMO Interacting Motifs: Structure and Function"

_cells, 2021, doi:10.3390/cells10112825_

Round 1
Reviewer 1 Report
Overview: This review is clearly written and provides a fairly up to date overview o SUMO-interacting motifs. As someone working in the field, I found it both interesting and useful to read. Therefore, I see this as a useful review for not only those working in the area of SUMO/SIM interactions, but also as a good entry point for those new to the field of research. I have two minor scientific points to make and a few editorial comments
Scientific comments:
On page 5 starting on line 177. There is also a crystal structure of the phosphorylated Daxx SIM in complex with SUMO1 (See reference 32). In this structure, I think K37, K39 and K46 of SUMO1 interact with negatively charged residues. Consistent with what is written, all three lysine residues are found in SUMO1, SUMO2 and SUMO3.
Starting on Page 7 there is a detailed discussion on SUMO/SIM interactions in histone function. There is a recent review on this subject in Nucleic Acids Research by Ryu and Hochstrasser (https://pubmed.ncbi.nlm.nih.gov/33885816/). This review is more detailed and should be cited. In many ways this makes this section somewhat redundant, although I still think it is a useful part of this review.
There is one important subject the authors failed to address relative to SUMO/SIM interactions. Given the very general definition of what constitutes a SIM sequence, is possible that there are many sequences classified as SIMs that do not in fact function as SIMs in a cellular context. For example, I am not sure that the SIM in RAD51 is a true functioning SIM, because it is in a structured region and buried in the core of a domain fold. Therefore, it is not clear to me how it could possibly interact with a SUMO protein short of the protein first unfolding. Mutational analysis in such cases proves very little, since it will likely disrupt the fold of the protein and hence many functions will be disrupted not just binding to SUMO. It would be useful for the authors to point out in this review that it is still not clear whether or not many sequences that are now classified as SIMs truly function as SIMs especially when they are located in a folded region as opposed to an intrinsically disordered region.
Editorial comments:
General:
The authors frequently use the word “while”. I prefer to use the word while only when there is a time element involved. In many cases, I think it would be clearer to the reader to use either “although” or “whereas” as opposed to “while”.
In the Figure legends, I think it would help to have a space after the letters designated the specific panels before the text starts.
Check your use of hyphenations for consistency: de-methylation versus demethylation, de-acetylation versus deacetylation, di-methylation versus dimethylation etc….
Specific:
Page 2 line 72: Better “….chemical shift perturbation studies.”
Page 2 line 75: I would replace “often about” with “typically”.
Page 3 figure legend 1: It is better to use “negatively charged region” as opposed to “acidic regions”. In the main text, the authors use negatively charged and this is a more accurate description of the region
Page 4 line 110: Better “…. NMR spectroscopy,…”
Page 5 line 159: I think (but was not sure) this is plural and should be “are” as opposed to “is”.
Page 7: I would be consistent with the use of the term PML nuclear bodies, which is abbreviated as PML-NBs. In several places, the authors substitute PML bodies.
Page 9 line 365: “Co-REST” should be “CoREST” for consistency with definition on line 355 of the same page.
Page 9 line 373: Better “semi-synthetic”
Page 13: better “co-repressor”
Page 13 line 513: Delete “however”
Page 13 line 514: I could not follow the logic in the sentence starting with “JAZ6 ubiquitylation…” . It would be helpful to make this clearer.
Reviewer 2 Report
The article of Yau et al. reviews the structure of SUMO-Interacting motifs and their involvement in different pathways. This is an area of intense research activity best illustrated by the number of structural, proteomics, and functional studies that were released in the last 5-10 years and that contributed to refining our understanding of SUMO-SIM interactions. The article of Yau et al. is overall well written but it suffers from several weaknesses.
Main points:
- Except for the involvement of SIMs in liquid phase transition, the article brings very little information on the exact involvement of SIMs in different functions. Sections 7 to 17 indeed present the involvement of SUMOylation in different pathways, but these examples bring little molecular or functional understanding except from the fact that SIMs are somehow involved. In their present form, these sections could easily be replaced with a table.
- The effects of the exact position of acidic/phosphorylated residues relative to the core SIM sequence could have been better described in view of available structural data. For example, there are now multiple SUMO-SIM structures containing acidic or phosphorylated residues immediately flanking the core SIM sequence and these structures suggest that not all positions contribute to productive binding to the same extent.
- The interplay between different types of post-translational modification that target the SUMO-SIM interaction could have been more developed in view of recent structural, biophysical, and proteomics data. For example, while different types of modifications are presented in section 4, the possible synergies or antagonisms could have been better developed.
- The last two points highlight one of the main weakness of the article: while several structures featuring SUMO-SIM interactions have been obtained in the last five years, the description by the authors is pretty similar to the description already available in old review articles (e.g. Kerscher O. SUMO junction-what's your function? New insights through SUMO-interacting motifs. EMBO Rep. 2007 Jun;8(6):550-5. doi: 10.1038/sj.embor.7400980. PMID: 17545995; PMCID: PMC2002525.).
- The authors should also remain cautious of sequences that were only shown to bind SUMO in isolation as these may not represent bona fide SIMs. For example, the SIM of Rad51 is located deep into the hydrophobic core of the protein (contrarily to what is shown in Figure 3D) and the observed defects upon mutation could be due to the misfolding of the protein. In this regard, it would be important that the author address the matter of SIMs accessibility. Structures can help in this regard.
- More general mechanisms implicating SUMO-SIM interactions such as the involvement of SIMs in protein group modification (Jentsch S, Psakhye I. Control of nuclear activities by substrate-selective and protein-group SUMOylation. Annu Rev Genet. 2013;47:167-86. doi: 10.1146/annurev-genet-111212-133453. Epub 2013 Aug 30. PMID: 24016193. and Psakhye I, Jentsch S. Protein group modification and synergy in the SUMO pathway as exemplified in DNA repair. Cell. 2012 Nov 9;151(4):807-820. doi: 10.1016/j.cell.2012.10.021. Epub 2012 Nov 1. PMID: 23122649.), one of the main hypotheses to explain the huge number of SUMOylations substrates relative to SUMO E3 ligases, could have been better discussed.
- Figure 1 and line 80: « An additional hydrophobic stretch is sometimes found on the opposite side 80 of the hydrophobic core relative to the negatively charged residues. » : What are the evidence that these additional hydrophobic stretch contribute to SUMO binding? While there are now several structures showing the involvement of the acidic sequences in binding, I don’t recall seeing data regarding these sequences and some of them are located pretty far from the SIM core (e.g. more than 10 residues away for DAXX, more than 40 for COI1).
- In section 3, the « binding energetics » and « thermodynamics » is almost absent. This is surprising given that multiple studies investigated SUMO-SIM interactions using techniques that provide thermodynamic data (e.g. microcalorimetry).
Minor points:
Figure 1: Suggestion to align the sequences according to the SIM core and indicate the position of the acidic residues in panel B.
Finally, please be aware that the hierarchy of the sections appears to have been lost during formatting (for example, section 8 at line 322 should be section 7.1.)
- Finally, I also felt that several articles should have been cited as they provide interesting information regarding SUMO-SIM interaction in general. For example:
* Jardin C, Horn AH, Sticht H. Binding properties of SUMO-interacting motifs (SIMs) in yeast. J Mol Model. 2015 Mar;21(3):50. doi: 10.1007/s00894-015-2597-1. Epub 2015 Feb 19. PMID: 25690366.
* Taupitz KF, Dörner W, Mootz HD. Covalent Capturing of Transient SUMO-SIM Interactions Using Unnatural Amino Acid Mutagenesis and Photocrosslinking. Chemistry. 2017 May 2;23(25):5978-5982. doi: 10.1002/chem.201605619. Epub 2017 Feb 14. PMID: 28121373.
* Newman HA, Meluh PB, Lu J, Vidal J, Carson C, Lagesse E, Gray JJ, Boeke JD, Matunis MJ. A high throughput mutagenic analysis of yeast sumo structure and function. PLoS Genet. 2017 Feb 6;13(2):e1006612. doi: 10.1371/journal.pgen.1006612. PMID: 28166236; PMCID: PMC5319795.
* Kötter A, Mootz HD, Heuer A. Insights into the Microscopic Structure of RNF4-SIM-SUMO Complexes from MD Simulations. Biophys J. 2020 Oct 20;119(8):1558-1567. doi: 10.1016/j.bpj.2020.09.003. Epub 2020 Sep 11. PMID: 32976759; PMCID: PMC7642298.
Round 2
Reviewer 2 Report
The authors made a good effort trying to address my concerns. The manuscript has improved, particularly in the structure section, and I now support the manuscript for publication. That said, several changes should be made prior to final publication:
Figure 1A. As discussed in the previous round, remove Rad51 from the list of SIMs.
Figure 1A. For PIASx and RanBP2, what are the evidence that the SIM core is composed of 5 residues?
Figure 1A. For PML, it appears that the acidic portion is provided by the phosphorylated serine residues located on the C-terminal part of the SIM core according to (Cappadocia L, Mascle XH, Bourdeau V, Tremblay-Belzile S, Chaker-Margot M, Lussier-Price M, Wada J, Sakaguchi K, Aubry M, Ferbeyre G, Omichinski JG. Structural and functional characterization of the phosphorylation-dependent interaction between PML and SUMO1. Structure. 2015 Jan 6;23(1):126-138. doi: 10.1016/j.str.2014.10.015. Epub 2014 Dec 11. PMID: 25497731.)
Figure 1A. For RNF4, suggestion to show the sequence of SIM2 (IVDL) instead of SIM3. This is because SIM2 is closer to consensus and displays a tighter binding to SUMO according to (Xu Y, Plechanovová A, Simpson P, Marchant J, Leidecker O, Kraatz S, Hay RT, Matthews SJ. Structural insight into SUMO chain recognition and manipulation by the ubiquitin ligase RNF4. Nat Commun. 2014 Jun 27;5:4217. doi: 10.1038/ncomms5217. PMID: 24969970; PMCID: PMC4083429.)
Figure 2B. Why are there 2 SUMOs depicted on residue K12 of histone H4?
Figure 3. In this figure, PCNA is depicted as poly-SUMOylated. What are the evidence for this poly-SUMOylation versus multi-SUMOylation of one or multiple PCNA subunits?
Finally, please add references to the following sentences:
Page 1: “SUMO contains an extended C-terminal region which is cleaved by a member of the ubiquitin-related protease family exposing a C-terminal di-glycine motif that is critical for covalent conjugation.” And the two following sentences.
Page 2 : “The SUMO acceptor lysine is often, though not always, embedded in a ѰKXE consensus motif, with Ѱ denoting a hydrophobic amino acid residue and X denoting any amino acid residue.”
